# Bio-Derived Carbon with Tailored Hierarchical Pore Structures and Ultra-High Specific Surface Area for Superior and Advanced Supercapacitors

**DOI:** 10.3390/nano12010027

**Published:** 2021-12-23

**Authors:** Fuming Zhang, Xiangshang Xiao, Dayakar Gandla, Zhaoxi Liu, Daniel Q. Tan, Yair Ein-Eli

**Affiliations:** 1Department of Materials Science and Engineering, Guangdong Technion-Israel Institute of Technology, 241 Daxue Road, Jinping District, Shantou 515063, China; fuming.zhang@gtiit.edu.cn (F.Z.); xiao.xiangshang@gtiit.edu.cn (X.X.); dayakar.gandla@gtiit.edu.cn (D.G.); zhaoxi.liu@gtiit.edu.cn (Z.L.); 2Department of Materials Science and Engineering, Technion-Israel Institute of Technology, Haifa 3200003, Israel; 3Grand Technion Energy Program (GTEP), Technion-Israel Institute of Technology, Haifa 3200003, Israel

**Keywords:** Metaplexis Japonica, carbon from biomass, electrolyte ion size, ionic liquid, supercapacitor

## Abstract

We report here on a hollow-fiber hierarchical porous carbon exhibiting an ultra-high specific surface area, synthesized by a facile method of carbonization and activation, using the Metaplexis Japonica (MJ) shell. The Metaplexis Japonica-based activated carbon demonstrated a very high specific surface area of 3635 m^2^ g^−1^. Correspondingly, the derived carbonaceous material delivers an ultra-high capacitance and superb cycle life in an alkaline electrolyte. The pore-ion size compatibility is optimized using tailored hierarchical porous carbon and different ion sized organic electrolytes. In ionic liquids nonaqueous based electrolytes we tailored the MJ carbon pore structure to the electrolyte ion size. The corresponding supercapacitor shows a superior rate performance and low impedance, and the device records specific energy and specific power densities as high as 76 Wh kg^−1^ and 6521 W kg^−1^, as well as a pronounced cycling durability in the ionic liquid electrolytes. Overall, we suggest a protocol for promising carbonaceous electrode materials enabling superior supercapacitors performance.

## 1. Introduction

Energy storage devices with high energy density and high-power density play an essential role in the rapidly growing power market [1,2,3,4,5]. Among these, supercapacitors have already found wide applications as a power source for auxiliary emergency power supplies, high-power supplies, start-stop power supplies for new energy electric vehicles, energy recovery systems, and smart grids [5,6,7]. However, supercapacitors suffer from a low energy density and have a relatively low market share in commercial electrochemical energy storage devices. Electrode materials play a significant role in boosting the capacitive performance of supercapacitors. Therefore, next-generation supercapacitors critically rely on advanced renewable, low-cost, eco-friendly electrode materials, possessing high adsorption, rapid ion/electron transport, and tunable surface chemistry [8,9]. In recent years, biomass-derived carbon electrode materials became significant and quite noticeable for being natural, uniform, and holing a precise template for electrode conversion. A great deal of work tempted biomass-derived porous carbon for high specific surface area (SSA), high porosity, rich pore structure, excellent physical and chemical stability, and good electrical conductivity favorable for electrochemical capacitors [10,11,12,13,14,15,16]. The electrochemical properties of biomass carbon greatly rely on biomass precursors’ structure, morphology, composition, and preparation methods. On the other hand, the commercial activated carbons usually manifest microporous characteristics with a narrow pore size ranging from 0.5 to 1.1 nm [17,18]. This type of tortuous micropores severely blocks the transport of ions, especially sizeable ions in the organic electrolyte and ionic liquids, leading to a low power and energy density at high operation current density.

To overcome the limitation in the conventional activated carbon, some researchers attempted hierarchical porous carbon via biomass precursors. Its macropores can act as ion-buffering reservoirs in the interior of carbon materials, mesopores can reserve as ion transport channels, and micropores as a critical location for charge storage [19]. The formation of the pore structures is generally dependent on the processing conditions of carbonization and activation. Pyrolysis carbonization is usually carried out at higher temperatures in an inert atmosphere, while hydrothermal carbonization in a closed container at lower temperature and higher pressure. As a pore-forming process, the activation’s temperature, time, atmosphere, and flow rate significantly affect the microstructure and properties of biomass-derived carbon [20,21,22,23]. In addition, specific surface area, heteroatom doping, graphitization degree, defects, and morphology of activated carbon also affect its electrochemical performance [24]. Shang and co-workers proposed conventional pyrolysis carbonization and KOH activation agents for walnut shell-based activated carbon (ACWS) with the highest specific surface area of 3577 m^2^ g^−1^ up to date [25]. The tunable pore structure assists packaged supercapacitor devices with ultra-high power and energy densities of 100 kW kg^−1^ and 120 Wh kg^−1^. Tour’s group proposed a facile and cost-effective approach to electrodes made from nitrogen-doped carbonized cotton used in flexible supercapacitor [26]. This strategy increased the wettability and additional faradaic pseudo-capacitance of the electrode material. In addition to the traditional process, the Li group developed a one-step route to synthesize the hierarchical porous carbon using calcium carbonate (CaCO_3_) as template and potassium oxalate (KC_2_O_2_) as activation agent [27]. The increased mesopores and graphitization degree greatly contribute to a rapid ion transport, low internal resistance, high capacitance, and energy density. So far, many natural waste materials have been attempted for supercapacitor application, including fruit and vegetable peels [28], willow catkins [29], peanut shell [30], watermelon [31], bamboo [32], soybean [33], lotus seedpods [34], metaplexis shell [35].

Yet, the pathway to utilize the SSA advantage and hierarchical pore structure promised by biomass for even higher energy density, and power density remains unclear. Matching properly selected pore size and electrolytes with larger ions can lead to the discovery of working partners that would effectively maximize the supercapacitor performance. Kondrat and co-workers found that optimizing the pore size of nanoporous carbon electrodes resulted in the maximal energy density using grand-canonical Monte Carlo simulations [36]. The use of large ions increased the operating voltages of the supercapacitor cells. Lian et al. and Nav Nidhi Rajput proposed a modified BSK model for porous electrodes to study the role of the pore size and ion adsorption on cell performance [37,38]. Yet, the ideal monodispersed porous carbon is to be experimentally developed.

Herein, unlike most of the porous carbons (large particles, cage, 2D platelet), we developed a porous carbon with hollow fiber shape by a facile activation method using Metaplexis Japonica husk as the precursor, which exhibits a world record high SSA of 3635 m^2^ g^−1^. The Metaplexis Japonica is widespread in the northern China, which can be collected in large quantities and manufactured into activated carbon, as one of the cost saving and environmentally friendly materials. We evaluated the compatibility of the hierarchical pore size with properly sized ions in three different electrolytes. The best performing MJ-5 electrode using an ionic liquid reaches the highest energy density. This porous carbon has a hollow fiber channel structure hosting both microporous and mesoporous structures. This remarkable structure provides plenty of locations for charge storage and accelerated ion transport. The supercapacitor cell exhibits high specific capacitance about 423 F g^−1^ at 5 mA cm^−2^ in 6 M KOH electrolyte, excellent kinetic behavior in the organic electrolyte, the high energy density of 76 Wh kg^−1^, and power density of 6521 W kg^−1^ under a working voltage of 3.5 V in ionic liquid electrolyte.

## 2. Experimental

### 2.1. Preparation of Activated Carbon (MJ-x)

Metaplexis Japonica (MJ) was collected in the mountains of Liaoning province in northeast China. The MJ was washed several times and dried at 70 °C in an oven, then crushed to small pieces. The MJ shell-based porous carbons were prepared by pre-carbonization and chemical activation using MJ shell as precursor and KOH as activation agent, respectively. Initially, the dried MJ pieces were pre-carbonized in a tube furnace at 500 °C under Ar protection for 1 h. Subsequently, the obtained chars (CMJ) were mixed with potassium hydroxide (KOH) solution in different KOH/CMJ mass ratios of 1:1, 3:1, 5:1. After drying at 80 °C, the mixtures were heat-treated at 400 °C for 1.5 h first and then at 800 °C for 2 h in Ar flow with a heating rate of 5 °C min^−1^. Then the resulting solid residue was washed with 1 mol/L HCl and abundant deionized water until the neutral pH value. The final drying was carried out at 100 °C for 24 h to obtain the MJ-x activated carbon, where x refers to the weight ratio of solid KOH to CMJ.

### 2.2. Microstructure Characterization

Scanning electron microscopy (SEM) was performed on a ZEISS Gemini 450 field emission, and transmission electron microscopy (TEM) was conducted by a JEM2100 instrument at an acceleration voltage of 200 kV. X-ray diffraction (XRD) and Raman spectra of MJ-x and YP50 were recorded by Smartlab 9 X-ray diffractometer at a scan rate of 6°/min using 150 mA current, 40 kV voltage, and copper target and a Renishaw-in Via Raman spectroscope (532 nm laser), respectively. Quantachrome Autosorb-iQ2-MP (Quantachrome Inst, Boynton Beach, FL, USA) and nitrogen isotherms were used to test specific surface area (SSA) and pore size distribution (PSD) by BET tests with degassing at 250 °C for 3 h for each 36 mg sample. The element distribution and functional changes were explored using X-ray photoelectron spectroscopy of Thermo Scientific ESCALAB 250Xi (Thermo Fisher, Waltham, MA, USA) and monochromatic Al target.

### 2.3. Electrochemical Measurements

The carbon electrode was fabricated by mixing 82% as-prepared activated carbon, 10% carbon black (Super Li), and 8% binder of PVDF. Then a vacuum mixer was used for 10 min to achieve a homogeneously mixed carbon slurry by adding solution NMP. The slurry was followed by coating on a 1 cm × 1 cm nickel foam an aluminum foil collector then dried in a vacuum oven overnight at 100 °C and 120 °C, respectively. The electrochemical tests were evaluated by assembling three-electrode system (Hg/HgO as the reference and Pt plate as the counter electrodes). The electrodes for organic SCs were prepared by homogeneously mixing 82 wt% activated carbon powder, 10 wt% carbon black (Super Li) as the conductor, and 8 wt% poly(tetrafluoroethylene) (PTFE) as the binder. The mixture was stirred to a sticky state followed by pressing at 15 MPa on the carbon-coated aluminum foil, assembling symmetrical EDLC supercapacitor device in a CR2032 coin cell. The loading mass is about 12~13 mg. The test voltage ranges from −1 V to 0 V for 6 M KOH electrolyte, 0–2.7 V for organic electrolyte, and 0–3.5 V for the ionic liquid electrolyte. The electrolytes having different ion sizes are *N*,*N*-dimethylpyrrolidine tetrafluoroborate ammonium (DMPBF_4_, provided by Tiangong University, Tianjin, China), spiro-(1,1′)-bipyrrolidinium tetrafluoroborate ammonium (SBPBF_4_, Jiangsu Guotai International Group Guomao Co. Ltd., Zhangjiagang, China), Tetraethyl ammonium tetrafluoroborate (TEABF_4_, Macklin, Shanghai, China). The ionic liquid is EMIBF_4_ purchased from Macklin, Shanghai, China.

The electrochemical tests were conducted using the Gamry electrochemical workstation (Interface 1010E, Gamry, Warminster, PA, USA) and Arbin Station (Arbin System, College Station, TX, USA). CV measurement was performed at different scan rates from 5 to 100 mV s^−1^ for the KOH system and from 0.2 A g^−1^ to 5 A g^−1^ for the organic system. The specific capacitance values in aqueous electrolytes were calculated using GCD curves from the following Equation [29]:$$C=\frac{I\times \Delta t}{\Delta V\times m}$$
where *C* denotes the specific capacitance (F g^−1^) of the electrode materials, *I* represents the discharge current (*A*), Δ*t* is the discharge time (s), m is the mass(g) of active material loaded, and Δ*V* denotes the potential window (*V*) excluding IR drop.

The gravimetric capacitances of a single electrode for organic and ionic liquid electrolytes were calculated according to the Equation [25]:$$C_{sp}=\frac{2I\times \Delta t}{\Delta V\times m}$$
where *I* (*A*) is the current, *t* (*s*) refers to the discharge time, Δ*V* (*V*) is the voltage window, and *m* (*g*) represents the weight of active materials in the single working electrode.

The energy and power density was calculated by the following Equations [25], respectively:$$E=\frac{C_{sp}\times \Delta V^{2}}{8\times 3.6}$$
$$P=\frac{E}{t}\times 3600$$
where *C* is the double electrode specific capacitance, *E* (Wh kg^−1^) is the specific energy density, and *P* (W kg^−1^) is the specific power density of a final supercapacitor device.

Electrochemical impedance spectroscopy (EIS) was performed at open circuit voltage within the 10–0.01 Hz frequency range. The imaginary part of the capacitance, *C*″ (F g^−1^) was calculated by means of the Equation [25]:$$C\prime \prime =\frac{\left|Re(z)\right|}{2\pi fm\left[{\left|Im(z)\right|}^{2}+{\left|Re(z)\right|}^{2}\right]}$$
where *f* is the frequency in Hz, and *Im*(*Z*) and *Re*(*Z*) are the imaginary and real parts of the impedance (*Ohm*), respectively. The term *τ*_0_ refers to the relaxation time constant estimated from the frequency *f*_0_ by using the formula *τ*_0_ = 1/*f*_0_, in which *f*_0_ is the peak frequency in the imaginary capacitance plot.

## 3. Results and Discussion

### 3.1. Microstructure and Chemical Analyses of the Activated Carbonaceous Materials

The conceptual diagram of the detailed preparation process of MJ-biomass porous carbon is shown in Figure 1. The process has been described in the experimental section. After the sample way of annealing and activating, we obtained a hollow fiber with different pore structures.

The morphology and microstructure of precursor were shown in Appendix A. Abundant fiber structures and pleated surfaces are captured in Appendix A. SEM images and local magnification in Appendix A show that the biomass generally has an original hollow fiber structure with a wall thickness of 1~2 μm. After KOH activation (Figure 2a–c), some large pore structures were observed, but the original structure was not completely collapsed; instead, the initial shape is maintained due to the thick wall. The MJ-5 SEM image at a lower magnification also shows that most of the activated carbon possess hollow fiber structures although it is not an intact structure and somewhere is broken or chipped. The zoom-in image for MJ-5 reveals an impressive image with a hierarchical pore structure on the fiber wall in the range of 0–400 nm (Figure 2f). In contrast, MJ-1 and MJ-3 presented in Figure 2d,e possess fewer holes on the fiber surface. MJ biomass-derived porous carbon structure diagrams are presented in Figure 2g. MJ-5 shows a hierarchical pore structure consisting of macropores, mesopores, and micropores. Among them, the macropores can act as an ion buffering reservoir; the mesopores, as fast transport channels, play a critical role in the ion diffusion, especially for the larger ion; the micropores can offer charge storage location [39]. Figure 2h schematically shows a more detailed structural explanation.

In order to further investigate the microstructure of MJ-x, the XRD, Raman, and XPS analyses were carried out, as shown in Figure 3. The two typical broad peaks at around 2θ = 23.4° and 43.5° in Figure 3a are assigned to the (002) and (101) reflection peaks of amorphous carbon, implying that the MJ-x activated carbon possess a low degree of graphitization. The Raman spectroscopic response was conducted to analyze the MJ-x’s specific structure further, as shown in Figure 3b. Two distinct characteristic peak bands, after fitting, situated at around 1338 and 1596 cm^−1^ are ascribed to the D band (defects and disorder) and G band (graphitic carbon), respectively. The intensity ratio of D and G (ID/IG) further expounds the graphitic and defective degree. A obtained high ID/IG value manifests more defects in the carbon products [40]. The ID/IG values of MJ-1, MJ-3, MJ-5, and YP-50 are 0.90, 0.92, 1.27, and 0.97, respectively, suggesting more defects being induced in the carbon with increasing KOH dosage, which is favorable for providing energy storage sites. To further explore the porosity of MJ-x products, the N_2_ adsorption-desorption isotherms and pore size distributions of all samples were conducted as shown in Figure 3c,d. The isotherms of all samples displayed relatively high nitrogen uptake at low relative pressures, indicating the presence of a great number of micropores. Compared with MJ-1 and YP-50 samples, at moderate relative pressures (P/P_0_ = 0.4–1.0), there is a clear hysteresis loop (IV-type isotherms) for MJ-3 and MJ-5, elucidating a great amount of mesopores generated. Moreover, a slight upward tendency at relatively high pressure (P/P_0_ = 0.95–1.0) reveals the existence of macropores [41,42]. Furthermore, the pore size distribution as shown in Figure 3c obtained using DFT method mode also confirms that the as-is carbon covers micropores (<2 nm) and mesopores (2–6 nm). The calculated SSA of MJ-5 based on the BET test is up to 3635 m^2^ g^−1^, which is higher than YP-50 and most other reported carbon materials. Additionally, the proportion of mesopores for MJ-5 (15.4%) is also higher than that of YP-50 (5.4%). The ultra-high SSA result is likely attributed to the synergistic effect of 3D hollow fiber structure with thick wall and KOH activation. More details about pore characteristics of as-prepared MJ-x are exhibited in Appendix A.

Besides, elemental analysis was carried out to estimate the chemical composition of MJ-x, as presented in Appendix A. It is clearly shown that the increase of KOH dosage increases the C content of MJ-1, MJ-3, and MJ-5 from 83.7% to 86.5% and 92.2% and decreases the oxygen concentrations gradually from 16.3 wt% to 13.5 wt% and eventually to 7.5 wt%, respectively. Compared with the Raman results, the decrease in heteroatom contents was conducive to producing defects and facilitating the transformation of the related functional groups into pores [43,44]. The chemical states and groups of MJ-x and YP-50 carbons were further measured using XPS. The XPS survey spectra (Figure 3e) displayed two peaks at 284.8 eV (C1s) and 532.6 eV (O1s). The fitted O1s peaks at 531.8 eV, 532.6 eV, and 533.5 eV in Figure 3f further demonstrate the existence of quinone and ketone groups (C=O), phenol groups (C-OH), and/or ether groups (C-O-C), respectively. The further analysis of C1s high-resolution spectrum as displayed in Appendix A can be convoluted into four peaks: C-C (284.8 eV), C-O (285.9 eV), C=O (286.8 eV), and -O-C=O (289.1 eV). Particularly, quinone/ketone oxygen, carbonyl, and ether oxygen serve as important oxygen functionalities to improve the surface wettability, while quinone oxygen could contribute to pseudo-capacitance [45,46].

### 3.2. Electrochemical Behavior of MJ-x Carbonaceous Materials in Aqueous Electrolyte

The electrochemical performances of MJ-x and comparison with prior data were first assessed using a three-electrode system in a 6 M KOH aqueous electrolyte, as shown in Figure 4 [47]. A comparison of CV curves at 10 mVs^−1^ for different samples is presented in Figure 4a. MJ-5 delivers the greatest curve area, mainly due to the rich micropores; the small K^+^ can access the inside of pores, exhibiting around two times larger area than that of a conventional YP-50 sample. Similarly, the GCD plots at 5 mV s^−1^ for all samples, as shown in Figure 4b display the same trend. MJ-5 has symmetrically triangular profiles with a long discharging time. The detailed CV and GCD plots at different scan rates from 5 mV s^−1^ to 100 mV s^−1^ and current densities from 5 mA cm^−2^ to 100 mA cm^−2^ is presented in Appendix A. An apparent increase in specific capacitance of MJ-5 at different current density (from 5 mA cm^−2^ to 100 mA cm^−2^) are displayed in Figure 4c. MJ-5 delivers a high specific capacitance about 423 F g^−1^ at 5 mA cm^−2^ and 328 F g^−1^ at 100 mA cm^−2^, while MJ-1 and YP-50 only possess 144 F g^−1^ and 106 F g^−1^ at 5mA cm^−2^, respectively, which was significantly higher than most of the reported works as displayed in Figure 5 [48,49,50,51,52,53,54,55,56,57,58,59]. The lower proportion of mesopores and micropores for MJ-1 and YP-50 leads to an inferior rate charge storage. 

Additionally, Figure 4d shows the steep and linear Nyquist impedance plots of all samples at low frequencies and small semicircles at high frequencies. In comparison with MJ-1, 3 and YP-50, MJ-5 shows the smallest Rst and Rct, demonstrating the ideal capacitive behavior. Furthermore, the normalized imaginary part of the capacitance versus frequency plots (Figure 4e) determines the relaxation time constant (*τ*_0_), which reflects the rate of ion adsorption and desorption. Compared with MJ-5 and YP-50 based on a three-electrode system, MJ-5 exhibits lower *τ*_0_ = 4.69 s and higher imaginary capacitance than YP-50. The relaxation time constant defines the boundary line between capacitive (at frequencies below 1/*τ*_0_) and predominant resistive (at frequencies above 1/*τ*_0_) behavior with varying frequencies [47].

The ideal capacitive behavior results from better intrinsic electronic properties and superior efficiency of ion transport in the unique electrode network compared with YP-50. At a higher current density (100 mA cm^−2^), the MJ-5 shows stable cyclic stability, which retains 100% of its initial capacitance after 10,000 cycles. In summary, the unique hollow fiber hierarchical porous structure is favorable for the liquid accessibility to the micropores. It promotes ion transport in the mesopores’ matrix effectively, thus enhancing the capacitance and rate capability in an aqueous system.

### 3.3. Electrochemical Behavior MJ-x Carbonaceous Materials in Organic Electrolyte

Mesopores in porous carbon can accommodate relatively high loading of ions from electrolytes while micropores cannot bear. This carbon morphology can lead to internal strain and decreased energy density with repeated charge/discharge cycles. The relatively large cation (Volume = 162.08 Å^3^) in conventional electrolyte (TEABF_4_) cannot readily access the abundant micropores of carbon electrode. Therefore, it is necessary to find out the matching ions suitable for the pores. Given the MJ-5 carbon electrode, we investigated three organic electrolytes with different cation ion sizes, TEABF_4_, SBPBF_4_, and DMPBF_4_, respectively (TEA^+^ > SBP^+^ > DMP^+^). The detailed size information is shown in Appendix A.

Figure 6a showed the CV curves at 100 mV s^−1^ for MJ-5 with disparate electrolytes. MJ-5 with DMPBF_4_ electrolyte results in the rectangular shape, while the other two electrolytes cause date pit shape, especially for the case of bigger ion TEABF_4_. This is mainly attributed to the accessibility of smaller ions to porous carbon. At a high scan rate, the smaller ions are much easier to adsorb and desorb through mesopores to micropores. Figure 6b shows the CV plots of electrodes with different levels of pores (MJ-1, MJ-3, MJ-5) matching DMPBF_4_ at a scan rate of 100 mV s^−1^. The multi-level pore structure electrode (MJ-5) demonstrated excellent capacitive behavior due to the effective pore-ion spaces. By contrast, the CV plots of all samples at different scan rates from 5 mV s^−1^ to 100 mV s^−1^ were shown in Appendix A. The MJ-5 with DMPBF_4_ possessed the best capacitive behavior, especially at high scan rates, suggesting the outstanding contribution to the capacitance of pore-ion size space. Furthermore, the effect of different electrolyte ion sizes and pore sizes on the capacitance was investigated in Figure 6c,d. Compared to SBP^+^ and TEA^+^, DMP^+^ based electrolyte working with MJ-5 displayed the longest discharge time. Moreover, based on DMPBF_4_, MJ-5′s hierarchical porous carbon structure exhibits the best discharge behavior. 

At the same condition, the rate capability has also been calculated by the GCD test with a current density ranging from 200 to 5000 mA g^−1^ as shown in Figure 6e,f. GCD plots of MJ-x in different organic electrolyte are shown in Appendix A. MJ-5 with DMPBF_4_ electrolyte possessed optimal rate performance. The specific capacitance obtained from the discharge time of the GCD curves can reach a high value of 127 F g^−1^ at a current density of 200 mA g^−1^ and 77 F g^−1^ at 5000 mA g^−1^ under 2.7 V, respectively. For MJ-3 and MJ-1, the specific capacitance can reach 123, 122 F g^−1^ at 200 mA g^−1^ and 67, 66 F g^−1^ at 5000 mA g^−1^, respectively. The preferable rate capability of MJ-5 is ascribed to the mesopores channel in the hollow fiber favorable for ion transport. Additionally, the distinction of specific capacitance for the MJ electrodes having different pore sizes based on the same electrolyte was shown in Figure 6f. The partnering of MJ-5 and DMPBF_4_ performs the best. Accordingly, the impedance R_ct_ of MJ-5 in DMP^+^ based electrolyte device is much smaller than the other samples regardless of different electrolytes or different pore sizes in carbon electrode, as shown in Figure 6g,h. These differences exhibit a higher ion diffusion efficiency that is also consistent with the result of the three-electrode system. A schematic of an EDLC cell is shown in Figure 6i where large ions can access the mesopores in the hollow fiber of MJ-5 electrodes. When testing the EDLC cell stability in different electrolytes, outstanding capacitance retention of 86% remains over 10,000 cycles at a current density of 5 A g^−1^ (Figure 6j), demonstrating its superior electrochemical durability.

To sum up, these results demonstrated the synergistic effect of suitable ion size and hierarchical porous network, which contributes to the superior energy storage property. The relationship between different ion size and hierarchical porous carbon, are shown in Figure 7. First, multi-level ion-matching spaces at <1 nm maximize the large surface area (3635 m^2^ g^−1^) as the charge storage location. Second, the DMP^+^ with a smaller Van der Waals Volume of 117.29 Å^3^ (ion size of 6 Å (height)) than SBP^+^ (143.3 Å^3^) and TEA^+^ (162.08 Å^3^) can enter the mesopores channel ranged in 2–6 nm more effectively. The sufficient mesopores provide an efficient and reversible charge pathway for improving the kinetic behavior, especially for smaller ions at high current density. Third, the unique straight-hollow fiber structure functions as an ion buffer reservoir prevent ion stacking during transport. Therefore, the complex microstructure in the high area carbon electrode requires consideration of many external factors contributing to the capacitance. The high pore-ion size compatibility plays a critical role in the charge storage of hierarchical porous carbon.

### 3.4. Electrochemical Behavior of MJ-5 Carbonaceous Material in Ionic Liquid Electrolyte

To further improve the energy density of the EDLC device, we studied the behavior of the MJ-5 electrode using an ionic liquid electrolyte (1 M EMIBF_4_ in ACN) by a series of electrochemistry characterization. The CV curves of MJ-5 at 3.5 V from 5 mV s^−1^ to 100 mV s^−1^ shown in Figure 8a maintain a rectangular shape even at a high scan rate of 100 mV s^−1^, indicating an excellent capacitive behavior. While the CV cures of YP-50 exhibit inferior shape with increasing scan rate, as shown in Appendix A. The GCD plots present a symmetric triangle shape, as shown in Figure 8b, delivering a higher discharge time than YP-50 (Appendix A). Moreover, the self-discharge behavior of the EDLC cell based on MJ-5 and YP-50 using EMIBF_4_/ACN electrolyte was studied under an open-circuit voltage, as shown in Figure 8c. The cell with MJ-5 retains a voltage of 1.28 V compared to 0.46 V for the cell with YP-50 after 13 h of self-discharge. The result indicates that a high surface area carbon electrode with efficient mesopores is favorable for ion accessibility and can host more charges than YP-50 with limited mesopores.

Their impedance spectra after CV and GCD tests are also measured and shown in Figure 8d. The semicircle diameter represents the resistance generated by the charge transfer resistance (R_ct_) between the electrolyte and electrode interface [59]. R_ct_ of MJ-5 (19.2 Ω) turns out to be lower than the cell with YP-50 (29.6 Ω), manifesting a fast ion transport. The specific capacitance of two-electrode cell using ionic liquid is shown in Figure 8e. Impressively, the MJ-5 delivered a superior rate performance from 0.2 A g^−1^ to 5.0 A g^−1^ with the corresponding specific capacitance of 164 F g^−1^ and 90 F g^−1^, respectively. The energy density and power density are calculated at different current density as shown in the Ragone-like plot in Figure 8f [60,61,62,63,64,65,66,67,68,69]. The optimal energy density and power density of MJ-5 can reach 76 Wh kg^−1^ and 6521 W kg^−1^, which is higher than most reported results. Figure 8g shows the cyclic stability of MJ-5 and YP-50 under 3.5 V and a current density of 2.5 A g^−1^, where MJ-5 retains 84% of its initial capacitance after 5000 cycles, while YP-50 keeps only 74% after 4000 cycles. Overall, the unique pore structure and high porosity in the MJ-5 are desirable for high energy density when using an ionic liquid system.

## 4. Conclusions

A special type of biomass precursor is well converted to an activated carbon featuring a 3-dimensional hollow fiber comprising mesopores and micropores, exhibiting a highest SSA of 3635 m^2^ g^−1^. The hierarchical porous structure provides effective and sufficient spaces for tailoring pore-ion size compatibility and the use of ionic liquid electrolytes. Benefiting from the tunable pore size and hollow fiber channel, fast ion transport and diffusion become feasible for delivering a high specific capacitance of 423 F g^−1^ at 5 mA cm^−2^ in 6 M KOH electrolyte, and superior rate performance for small ion size electrolyte (DMPBF_4_). Moreover, the ideal pore-ion matching enables the corresponding EDLC cells to reach a high energy density of 76 Wh kg^−1^ and a high power density of 6521 w kg^−1^ in an ionic liquid electrolyte. Therefore, this work demonstrates that maximizing the utilization of pore size distribution and pore-ion compatibility is equally essential for the energy storage enhancement of supercapacitors. The Metaplexis Japonica-derived porous carbon is a promising electrode material for practical supercapacitor applications being more compatible with electrolytes.

## Figures and Tables

**Figure 1 nanomaterials-12-00027-f001:**
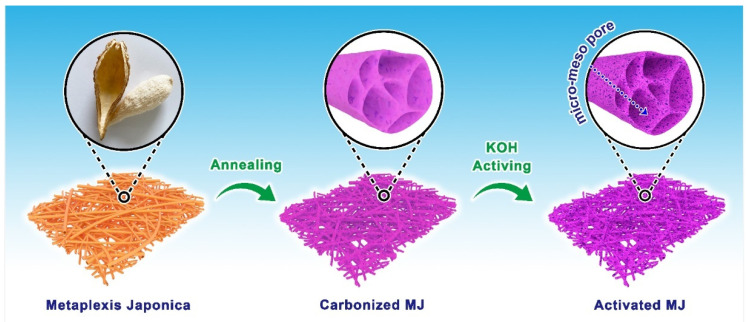
A conceptual diagram detailing the preparation process of MJ-biomass porous carbon.

**Figure 2 nanomaterials-12-00027-f002:**
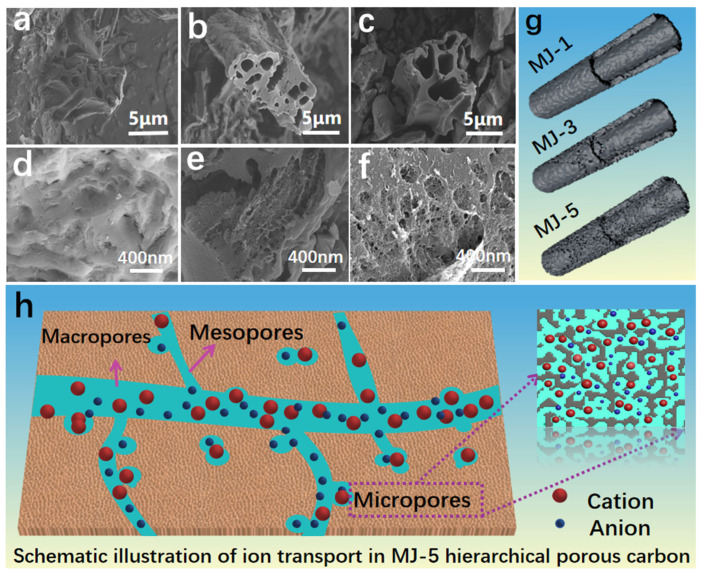
SEM images of activated carbon materials: (**a**) MJ-1; (**b**) MJ-3; (**c**) MJ-5; Corresponding local magnification; (**d**) MJ-1; (**e**) MJ-3; (**f**) MJ-5; (**g**) Structural schematic of MJ biomass-derived porous carbon; (**h**) Schematic illustration of ion transport in MJ-5 hierarchical porous carbon.

**Figure 3 nanomaterials-12-00027-f003:**
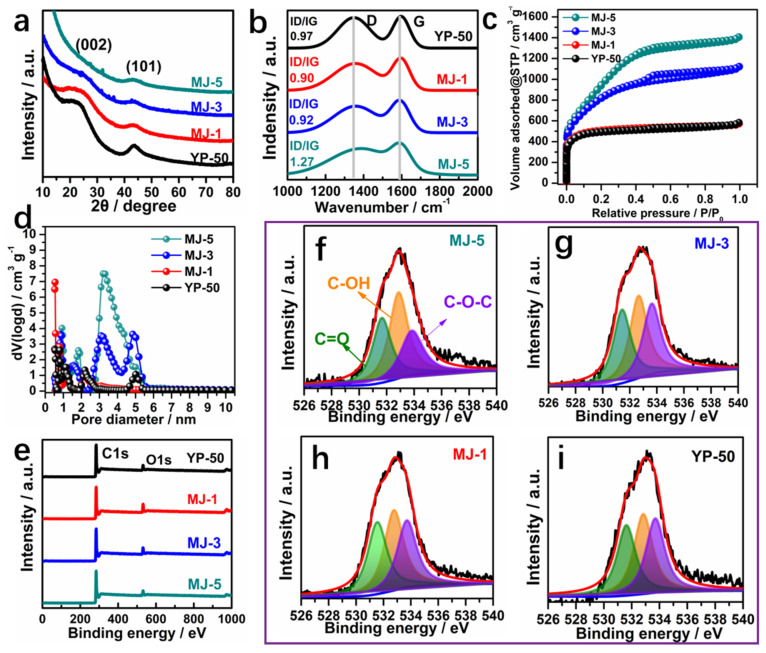
(**a**) Typical XRD patterns of MJ-x and YP-50; (**b**) Roman spectra of MJ-x and YP-50; (**c**) N_2_ adsorption-desorption isotherms of MJ-x and YP-50; (**d**) Pore size distributions of MJ-x and YP-50; (**e**) XPS survey spectra of MJ-x and YP-50; (**f**–**i**) High-resolution O1s spectra of MJ-x and YP-50.

**Figure 4 nanomaterials-12-00027-f004:**
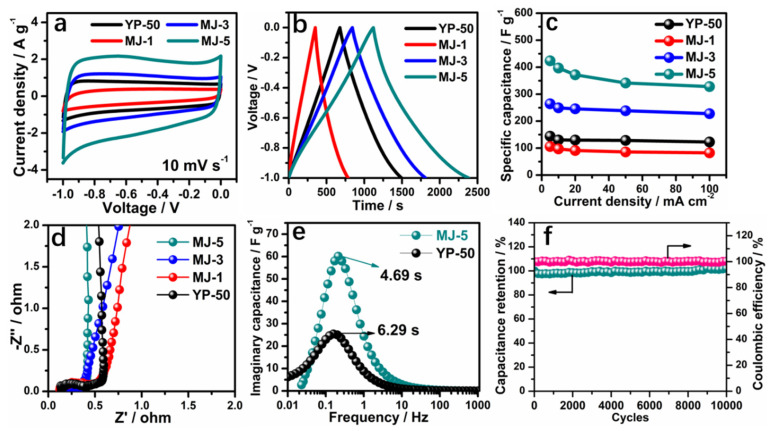
Electrochemical performance of MJ-x in 6 M KOH. (**a**) CV curves of MJ-x and YP-50 at a scan rate of 10 mV s^−1^; (**b**) GCD curves of MJ-x and YP-50 at 5 mA cm^−2^; (**c**) Rate performance of MJ-x and YP-50; (**d**) Nyquist plots of MJ-x and YP-50 [47]; (**e**) The normalized imaginary part of the capacitance versus frequency plots for MJ-5 and YP-50; (**f**) Cycling performance of MJ-5 at 500 mA cm^−2^.

**Figure 5 nanomaterials-12-00027-f005:**
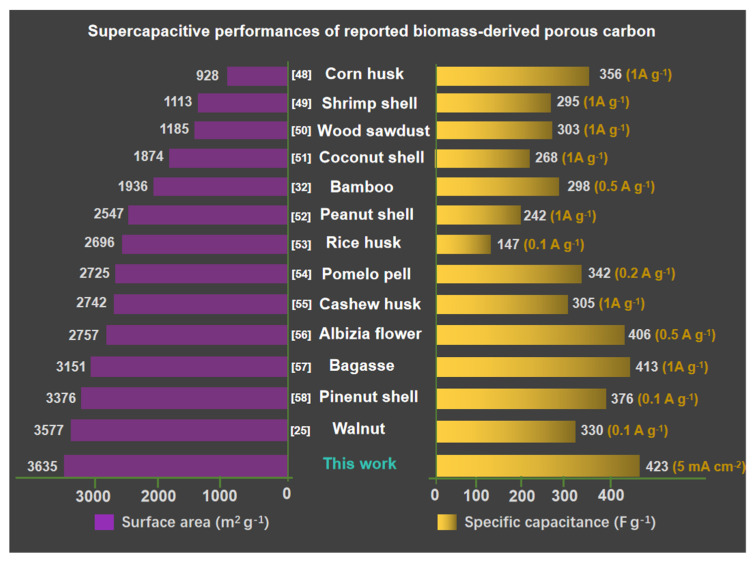
Comparison of the specific capacitance and surface areas of MJ-5 with the state-of-the-art biomass-derived carbon materials [48,49,50,51,52,53,54,55,56,57,58].

**Figure 6 nanomaterials-12-00027-f006:**
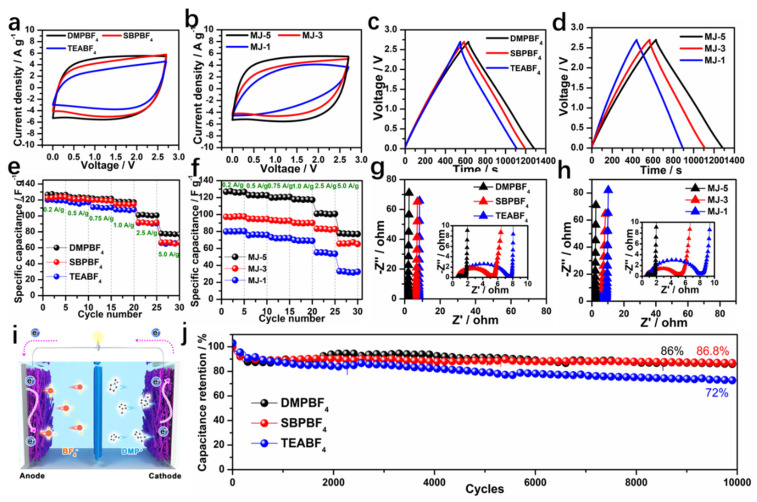
Electrochemical performance of MJ-x in organic electrolyte. (**a**) CV curves of MJ-5 using different electrolytes at 10 mV s^−1^; (**b**) CV curves of MJ-x using DMPBF_4_; (**c**) Specific capacitance of MJ-5 using different electrolytes at 0.2 A g^−1^; (**d**) GCD curves of MJ-x using DMPBF_4_ at 0.2 A g^−1^; (**e**) Rate performance of MJ-5 at different electrolyte; (**f**) Rate performance of MJ-x at DMPBF_4_; (**g**) Nyquist plots of MJ-5 using different electrolytes; (**h**) Nyquist plots of MJ-x using DMPBF_4_; (**i**) Schematic of a supercapacitor device; (**j**) Cycling performance of MJ-5 in different electrolytes.

**Figure 7 nanomaterials-12-00027-f007:**
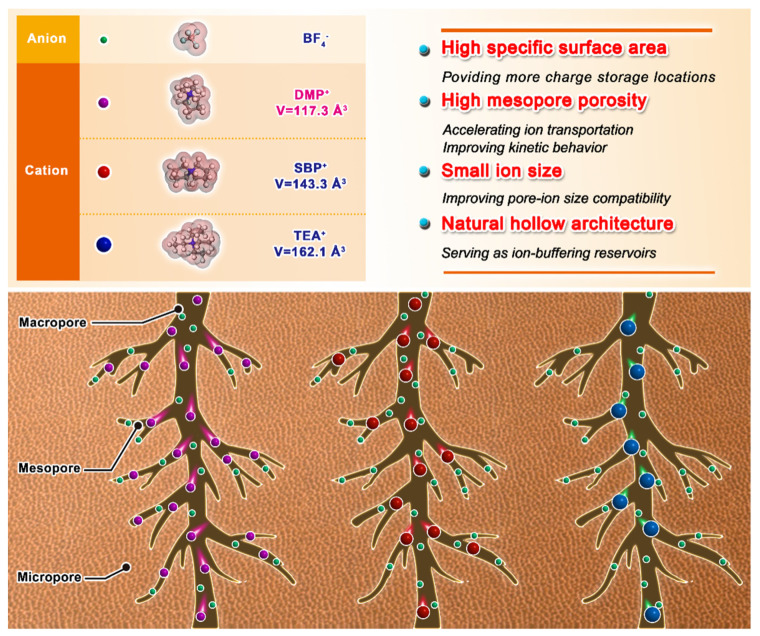
Schematic diagram of ion size of different organic electrolytes.

**Figure 8 nanomaterials-12-00027-f008:**
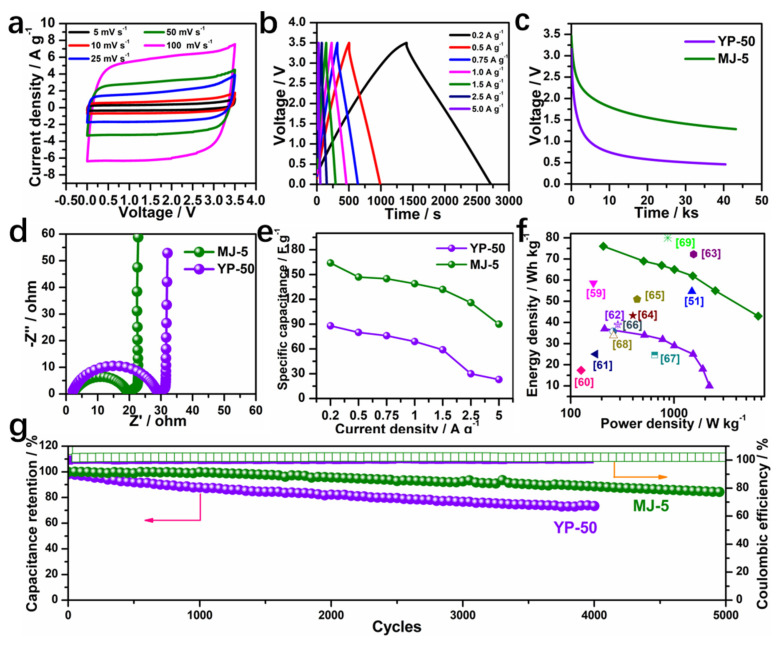
Electrochemical performance of MJ-5 in an ionic liquid electrolyte. (**a**) CV curves of MJ-5 from 5 mV s^−1^ to 100 mV s^−1^; (**b**) GCD plots of MJ-5 from 0.2 A g^−1^ to 5.0 A g^−1^; Comparative electrochemical performance of MJ-5 and YP-50; (**c**) Self-discharge; (**d**) Impedance; (**e**) Rate capacity; (**f**) Ragone plots compared with other reported results [60,61,62,63,64,65,66,67,68,69]; (**g**) Cycling performance at 2.5 A g^−1^.

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
