# Peer review of "Bio-Derived Carbon with Tailored Hierarchical Pore Structures and Ultra-High Specific Surface Area for Superior and Advanced Supercapacitors"

_nanomaterials, 2021, doi:10.3390/nano12010027_

Round 1

Reviewer 1 Report

The authors report on the Hollow-fiber Carbon with Tailored Hierarchical Pore Structures and Ultra-high Specific Surface Area for Superior and Advanced Supercapacitors . Though use of biomass derived carbon materials is not new however, the authors claim to derive the carbon from  Metaplexis Japonica . My first concern is about the tile of the paper that is somehow misleading. SEM images though show some hollow fibres but that depends on the fact from which part of the material these images are taken. To me they dont see uniform and only a part of the material shows these structures. Therefore, explaining all the results on the basis of  observed holllow structures is not correct. Although the performance of the material is good but the authors should describe the results and title generally instead of stressing on hollow structure. The authors may reproduce the carbon and see how the structure look like.

2: No references are provided for the equations used.

3: Two electrode study on the supercapacitors should be provided.

Author Response

attachment

Reviewer 2 Report

This article concerns the successful conversion of carbon biomass to amorphous carbon with a controlled pore size. The results are presented clearly, well illustrated with figures and reasonably discussed. Undoubtedly, the method, which the authors developed for obtaining an amorphous-carbon material with a record value of specific surface area and electrochemical conductance, is promising and successful. The article is well written and can be recommended for publication in the Nanomaterials.

However, certain comments should be made. This concerns the structural-element analytical part of the article. Such an analysis of sp2 amorphous carbon is a very complicated task, so that only recently there have appeared works that summarize the accumulated experience in this area and offer some algorithms of a successful approach. This concerns the structure and total elemental composition of the samples (J. Non-Cryst. Sol. 2019; 524:11960), analysis of their XPS (Full. Nanot. Carb. Nanostr. 2020; 28:1010) and Raman (Nanomaterials 2020; 10:2021) spectra. The application of these algorithms to the analysis of the studied samples will undoubtedly significantly enrich this part of the work, turning the standard analysis into an interesting study of a new material. In connection with the growing attention to the sp2 carbon materials derived from biomass, it seems strange that there are no references to the work of Tour’s group, for example Nature 2020 577, 647, in the list of cited literature. Making up for this deficiency will undoubtedly improve the overall impression of the article.

Author Response

attachment

Reviewer 3 Report

The manuscript proposes a new supercapacitor which demonstrated high efficiency and stability. The work is very interesting and well explained. On my opinion, authors focused more on pore volume and, despite having done surface chemical characterization, the role of the chemistry of the material on performance was not discussed.

An economic analysis would also be interesting.

The description on Pag. 10-11, of the material characterization is confusing, with the reference of figures of the main manuscript and mixed with supplementary figures.

Specific comment: Pag. 6, pH value instead of "PH value".

Author Response

attachment

Reviewer 4 Report

An precious attempt with encouraging results. To increase the scientific soundness, I suggest following minor revisions:

  1. Abstract: “…shell. The Metaplexis Japonica based activated carbon demonstrated a very high specific surface area of 3,635 m 2 g 1 . Correspondingly, the derived carbonaceous material delivers an ultra high capacitance of 423 F g 1 at a current density of 5 mA cm 2 and 100% capacitance retention at 100 mA cm 2 in an alkaline electrolyte.”

Those numbers, in particular in the abstract, would have a higher impact and value when compared to numbers of state-of-the-art systems in order to give the reader the ability to rate this achievement.

  1. Abstract: “records energy and power densities as high as 76 Wh kg 1 and 6,521 W kg 1”

You relate vs. mass (= gravimetric), hence it is “specific” energy and “specific” power.

Please revise throughout the manuscript according electrochemical definitions in e.g. 10.1007/s41061-018-0196-1

  1. Figure 6 j: y-axis is hidden

Author Response

attachment

Round 2

Reviewer 1 Report

The paper seems fine now and can be accpeted